# A SAR Ship Detection Method Based on Adversarial Training

**DOI:** 10.3390/s24134154

**Published:** 2024-06-26

**Authors:** Jianwei Li, Zhentao Yu, Jie Chen, Hao Jiang

**Affiliations:** Naval Submarine Academy, Qingdao 264001, China; lgm_jw@163.com (J.L.); grass2009@163.com (J.C.); jhao_ocean@163.com (H.J.)

**Keywords:** synthetic aperture radar (SAR), ship detection, deep learning, data generation, adversarial training, gradient descent

## Abstract

SAR (synthetic aperture radar) ship detection is a hot topic due to the breadth of its application. However, limited by the volume of the SAR image, the generalization ability of the detector is low, which makes it difficult to adapt to new scenes. Although many data augmentation methods—for example, clipping, pasting, and mixing—are used, the accuracy is improved little. In order to solve this problem, the adversarial training is used for data generation in this paper. Perturbation is added to the SAR image to generate new samples for training, and it can make the detector learn more abundant features and promote the robustness of the detector. By separating batch normalization between clean samples and disturbed images, the performance degradation on clean samples is avoided. By simultaneously perturbing and selecting large losses of classification and location, it can keep the detector adaptable to more confrontational samples. The optimization efficiency and results are improved through K-step average perturbation and one-step gradient descent. The experiments on different detectors show that the proposed method achieves 8%, 10%, and 17% AP (Average Precision) improvement on the SSDD, SAR-Ship-Dataset, and AIR-SARShip, compared to the traditional data augmentation methods.

## 1. Introduction

Synthetic aperture radar (SAR) is a kind of all-day and all-weather tool which can provide high-resolution images. It is widely used in civil and military areas. Among them, finding ships from SAR images is one of the most popular directions for this tool [1].

Traditionally, the Constant False Alarm Rate (CFAR) detector is usually adopted in this area [2]. It firstly models the pixel distribution of the SAR image, and then calculates the model parameters. The discrimination threshold is computed according to the false alarm rate and the target pixel value. Pixel values that are higher than the threshold will be regarded as ship pixels, and those lower than the threshold will be regarded as background. Finally, these high-amplitude pixels are extracted and merged into targets. In essence, CFAR is a kind of unsupervised method in machine learning view, and its performance must be worse than the supervised method.

In addition to CFAR, the machine-learning-based detectors are also used in SAR ship detection. Feature extracting and classifier designing are the two core steps, and they show advantages compared to CFAR, as they are strongly supervised. With the emergence of deep learning in 2012, and its excellent performance in various computer vision tasks, SAR ship detection also began to use deep learning. The beginning of this was the SSDD dataset and the great performance of deep learning shown in the following paper [3]. After that, a large number of researchers began to seek solutions from deep learning, and a lot of papers were published [4,5]. The deep learning models are data-hungry; thus, if the dataset is not big enough, the models may face the problem of over-fitting. This problem is serious, especially in the scenarios where it is difficult to collect data, for example, SAR ship detection.

Data augmentation is an important method to hold back over-fitting, which is usually used in general object detection. It can provide more diversified samples by changing the appearance of images, and can extract more information from them; thus, it can make the model more generalized. It can also be divided into basic and advanced data augmentations. The representative methods of the former are manipulation and erasing, while the representative methods of the latter are mixing and auto-augment. The above methods are based on experience, and the generalization and effectiveness of them are poor in general.

As we all know, deep learning models are highly susceptible to adversarial attack. Adding an imperceptible disturbance to the image can lead to a wrong prediction [6]. We think that if we use adversarial examples in the right way, they can improve the performance of the detector, as they can generate additional training images by adding noise, so that we can regard them as one way of data augmentation.

This can be realized by adversarial training. Adversarial training adds adversarial samples to the training set, making the model gradually adapt to adversarial attacks during training. In this process, the sample is mixed with some small perturbations, and then the deep learning models adapt to this change, thus becoming more robust [7].

Therefore, in order to improve the performance of the SAR ship detector, images generated by adversarial learning are used for data generation [8]. The clean and adversarial images are normalized separately to avoid their influence on the clean ones. Through the K-step gradient ascent and one-step descent method, the detector is trained finely. The high-quality adversarial images improve the classification and localization ability of the deep learning models. Experiments on several detectors and datasets are conducted, and they are shown to have great advantages over traditional data augmentation methods.

The achievements of the paper are as follows:(1)The adversarial learning is used in SAR ship detection, and the principles and formulas of adversarial learning are described in detail.(2)The optimization method based on separated batch normalization (BN), selecting largely from classification and localization losses, and the K-step gradient ascent with one-step gradient descent are used to solve the adversarial learning formulas.(3)The experiments on the classical detectors and public datasets are conducted, which show obvious superiority compared to the traditional data augmentation methods.

## 2. Methodology

An adversarial attack can make the deep learning models output a wrong prediction, which shows great risk to the application. However, if we add adversarial samples to the input and use them to update the parameters, the models can learn to resist them, and the robustness can be improved. This is the core idea of adversarial training, which can be used as a way of data generation. When conducting adversarial training, some small perturbations (random noise) will be mixed into the training data (changes are small, but may lead to the wrong prediction), and then the deep learning models will be adapted to such changes [9].

Suppose that the objective function of training is as follows:(1)$$\underset{\theta }{\arg\min}\left[E_{\left(x,y\right)\sim D}\left(L\left(\theta ,x,y\right)\right)\right]$$

Here, D represents the distribution of the input, L represents the loss function, θ represents the parameters of the model, x is the training data, and y is its corresponding real label [10]. 

The adversarial training is designed to determine the best parameters to minimize the maximum risk, as follows:(2)$$\underset{\theta }{\arg\min}\left[E_{\left(x,y\right)\sim D}\left(\underset{\varepsilon \in S}{\max}L\left(\theta ,x+\varepsilon ,y\right)\right)\right]$$

Here, ε is the interference, and S is its interference range. The purpose of ε is to maximize the loss function $L\left(\theta ,x+\varepsilon ,y\right)$, which means trying to make the models’ output a wrong prediction. ε should not be too big, and it is usually limited by norm, for example ${\left\|\varepsilon \right\|}_{F}\leq constant$. Every sample x can generate an adversarial sample x+ε. The training sample and label $\left(x+\varepsilon ,y\right)$ are used to update the parameters θ and minimize the loss function.

In the above equation, ‘max’ means that we need to find a group of adversarial samples with the maximum loss within the sample space. The outer ‘min’ function refers to the fact that, faced with the sample set of such data distribution, the expected loss of the model on the adversarial sample set should be minimized by updating the model parameters [11,12,13].

The min–max formula can improve the performance of adversarial examples. However, they cannot generalize well-to-clean images. This is because the adversarial examples and clean images have different underlying distributions, which prevents the network from accurately and efficiently extracting valuable features from these two domains. Therefore, we use their respective BNs during training. During the prediction, only the BN parameters of clean samples are used. The formula is as follows:(3)$$\underset{\theta }{\arg\min}\left[E_{\left(x,y\right)\sim D}\left(\left(L\left(\theta ,x,y\right)\right)+\underset{\varepsilon \in S}{\max}\;L\left(\theta ,x+\varepsilon ,y\right)\right)\right]$$

This training method, handling the variation in clean and adversarial samples alone, results in a better learning performance.

The loss function of object detection can be represented as follows:(4)$$L\left(\theta ,x,y\right)=L_{cls}\left(\theta ,x,y\right)+w\cdot L_{loc}\left(\theta ,x,y\right)$$

Here, w is used for balancing the two losses. It includes classification subtask $L_{cls}$ and localization subtask $L_{loc}$. Both of them can be attacked and result in wrong predictions. Therefore, the purpose can be accomplished by attacking one of the subtasks. In order to fully consider this point, the larger of the classification loss and localization loss is selected as the final loss. This method enables the model to adapt to more confrontational samples. 

For deep learning models, the outer ‘min’ is non-convex, and the inner ‘max’ is non-concave. The SGD can be used for the external minimization, and the PGD (Projected Gradient descent) can be used for the internal maximization. The usually used K-step PGD requires K forward and backward passes across the network, whereas standard SGD updates only require one pass [14,15,16,17]. Thus, the generation procedure of adversarial training can increase the running time by an order of magnitude. For improving the training speed and increasing the accuracy, the following steps are proposed.

The gradient g of loss function is accumulated in *K* steps, as follows:(5)$$g_{t}=g_{t-1}+\frac{1}{K}E_{\left(x,y\right)\sim D}[\nabla_{\theta }L\left(\theta ,x+\varepsilon ,y\right)]$$

It uses the average gradient in *K* iterations. The disturbance is updated by gradient ascent, as follows:(6)$$\varepsilon_{t}=\prod_{||\varepsilon |{|}_{F}\leq \mathrm{constant}}\left(\varepsilon_{t-1}+\alpha \cdot g_{adv}/||g_{adv}|{|}_{F}\right)$$

In which α is ascent step size and $g_{adv}=\nabla_{\varepsilon }L_{\theta }\left(\theta ,x+\varepsilon_{t-1},y\right)$.

Then, the parameters of deep learning models are updated by gradient descent, as follows:(7)$$\theta_{m}=\theta_{m-1}-\tau g_{K}$$

In which τ is the learning rate.

The above method can be used to optimize Formula (1) quickly and accurately.

The overall architecture of the proposed method is summarized in Figure 1.

## 3. Experimental Results

### 3.1. Experiment Setup and Baseline

The experiments use the Ubuntu system, Intel (R) Core (TM) i9-10900K processor, GeForce RTX 3080 GPU (NVIDIA Corporate, Santa Clara, CA, USA). The Pytorch framework is used. The initial learning rate is 0.001, the minimum learning rate is 0.00001, and the stepwise learning rate is adopted. The four representative deep learning-based detectors are chosen to conduct the experiments, and they are as follows: two-stage detector Faster R-CNN [18], one-stage detector YOLOV3 [19], focal loss-based detector RetinaNet [20], and the anchor-free detector CenterNet [21]. The experiments are conducted on the three public datasets, SSDD, SAR-Ship-Dataset [22], and AIR-SARShip [23]. The data augmentation methods of cutout, random erasing, mixup, cutmix, augmix, gridmask, mosic, and copy/paste are selected as the traditional method [24,25,26,27,28,29,30,31,32]. The AP50 is used as the evaluation indicator.

The deep learning-based detector usually adopts some data augmentation methods. For example, Faster R-CNN uses resizing and random flipping; YOLOV3 uses resizing, random flipping, and photo metric distortion; CenterNet uses the resizing and random flipping; and RetinaNet uses the resizing and random flipping. We call them the build-in data augmentation methods. The performances of them are regarded as the baseline in this letter, as shown in Table 1.

In Table 1, FR, RN, Y3, CN, and AVE are the abbreviations of Faster R-CNN, RetinaNet, YOLOV3, CenterNet, and average, respectively. In Table 2, Table 3 and Table 4, DAM refers to the data augmentation method.

### 3.2. Performance on Different Datasets

#### 3.2.1. Performance on SSDD

SSDD was first made public in 2017, and it introduced deep learning into the SAR ship detection in 2017. It has 1160 images and 2456 ships. It has many large sized ships, and the ships are densely arranged in some images. The samples of SSDD are shown in Figure 2.

The results of different data augmentation methods and detectors on SSDD are shown in Table 2.

For the traditional data augmentation methods, 32 experiments are conducted. Compared to the build-in methods, 11 of them were increased, with an average increase of 5%, and 21 of them were decreased, with an average decrease of 7.9%. The average AP50 of the 32 combination is 0.85. The average of the build-in data augmentation method in Table 1 is 0.89. This shows that the traditional data augmentation methods would decrease the performance of the detectors overall. Only the specific combination of the detector and data augmentation methods can boost the performance. For example, Faster R-CNN with augmix can boost the performance from 84% to 92% (with 8% improvement) on SSDD.

Compared to the mediocre ability of the traditional data augmentation method, the proposed adversarial training-based method can receive a huge improvement with all of the detectors. For example, it receives 0.94, 0.95, 0.89, and 0.95 AP50 on SSDD, respectively. The average value is 0.93. It receives an improvement of 0.04 and 0.08, compared to the build-in and traditional methods.

What is more, in SSDD, the number of images is small, and most of them are easy samples. The performance improvement of the proposed method is limited, compared to other datasets, as shown below.

#### 3.2.2. Performance on SAR-Ship-Dataset

Compared to SSDD, the SAR-Ship-Dataset has more samples. It has 43,819 images and 59,535 ships in total. They are from Gaofen-3 and Sentinel-1. Samples in it have diversity scales and conditions. Figure 3 shows some examples of this.

The results of different data augmentation methods and detectors on the SAR-Ship-Dataset are shown in Table 3. 

For the traditional data augmentation methods, 32 experiments are conducted. Compared to the build-in methods, 17 of them were increased, with an average AP increase of 2.9%, and 15 of them were decreased, with an average decrease of 7.2%. The average AP50 of the 32 combination is 0.83. The average of the build-in data augmentation method in Table 1 is 0.85. This shows that the traditional data augmentation methods would decrease the performance of the detectors overall. Only the specific combination of the detector and data augmentation methods can boost the performance. For example, YOLOV3 with mosic can boost the performance from 79% to 84% (with 5% improvement) on the SAR-Ship-Dataset.

Compared to the plain ability of the traditional data augmentation method, the proposed adversarial training-based method can receive a huge improvement with all of the detectors. For example, it receives 0.93, 0.94, 0.89, and 0.94 AP50 on the SAR-Ship-Dataset, respectively. The average value is 0.93. It receives an improvement of 0.08 and 0.10, compared to the build-in traditional methods.

From these experiments, we can find that the proposed adversarial training is effective on different detectors trained on the SAR-Ship-Dataset.

What is more, the SAR-Ship-Dataset has a big data volume, and there are a lot of hard samples. This is especially suitable for the proposed adversarial training-based data augmentation methods. This is because it can generate a lot of perturbing samples, and thus make the detectors robust to the hard samples in the SAR-Ship-Dataset. 

#### 3.2.3. Performance on AIR-SARShip

AIR-SARShip has 21 whole SAR images in total. Resolutions of them are 1 m and 3 m, and the image size of them is 3000 × 3000 pixels. The samples of AIR-SARShip are shown in Figure 4.

The results of different data augmentation methods and detectors on AIR-SARShip are shown in Table 4. 

For the traditional data augmentation methods, 32 experiments are conducted. Compared to the build-in methods, 14 of them were increased, with an average increase of 5.7%, and 18 of them were decreased, with an average decrease of 8.4%. The average of them is 0.72. The average of the build-in data augmentation method in Table 1 is 0.74. This shows that the traditional data augmentation methods would decrease the performance of the detectors overall. Only the specific combination of the detector and data augmentation methods can boost the performance. For example, YOLOV3 with random erasing can boost the performance from 46% to 84% (with 38% improvement) on AIR-SARShip.

Compared to the plain ability of the traditional data augmentation method, the proposed adversarial training-based method can receive a huge improvement with all of the detectors. For example, it receives 0.95, 0.93, 0.73, and 0.93 AP50 on AIR-SARShip, respectively. The average value is 0.89. It receives an improvement of 0.15 and 0.17, compared to the build-in and traditional methods.

The AIR-SARShip dataset is designed to detect ships in large scene. The ships in large-width SAR images are sparse. There are a large number of samples in the dataset that are difficult to detect. The detector has a large performance improvement space compared to other datasets. The adversarial training is particularly suitable for this situation.

### 3.3. Ablation Experiments

From the above experiments, we can find that the traditional data augmentation methods cannot always promote the detection accuracy, and, in some cases, it would even decrease the performance. However, the proposed method in this paper can promote the accuracy with all of the detectors and datasets. In order to investigate the principle of the proposed method further, we conduct the following experiments. The results are shown in Table 5. 

On SSDD, the plain adversarial training method can improve the AP50 from 0.85 to 0.88. The separate batch normalization, maxing, and optimization can also improve it to 0.89, 0.91, and 0.93. For the SAR-Ship-Dataset, the plain adversarial training method can improve the AP50 from 0.83 to 0.87. The separate BN, maxing, and optimization can also improve it to 0.89, 0.90, and 0.93. On AIR-SARShip, the plain adversarial training method can improve the AP50 from 0.72 to 0.81. The separate BN, maxing, and optimization can also improve it to 0.84, 0.86, and 0.89.

We can find that the plain adversarial training can receive a big improvement, especially on the AIR-SARShip-Dataset, as it includes a lot of hard samples. The separate BN can receive an improvement of 0.02 on average. This is due to the fact that the separate batch normalizations of clean samples and disturbing samples can avoid the interference of the clean samples. Thus, the overall performance is improved. The maxing can receive an improvement of 0.02 on average. This can make the model adapt to more confrontational samples. The proposed optimization method can make the model converge better and faster, which can also receive an improvement of 0.027 on average.

### 3.4. The Detection Results

The detected results of the proposed method on the SSDD, SAR-Ship-Dataset, and AIR-SARShip are displayed in Figure 5. We found that it can cope with both the easy and hard samples, which reveals that the adversarial training is efficient.

## 4. Conclusions

This paper proposes an adversarial learning-based data augmentation method to address the issues of insufficient data diversity and limited performance improvement of traditional methods. The proposed method generates adversarial images by appending small perturbations to the image, and achieves the generation of interference images and optimization of model parameters by modeling and solving the max–min problem. During the model solving process, clean samples and interfering images were normalized separately, and the larger interference of the localization and classification subtasks was selected. A multi-step average disturbance and single-step gradient update were used to increase the efficiency and accuracy of the adversarial learning. Experiments on three datasets and four detectors show that it has significant performance advantages compared to traditional methods.

## Figures and Tables

**Figure 1 sensors-24-04154-f001:**
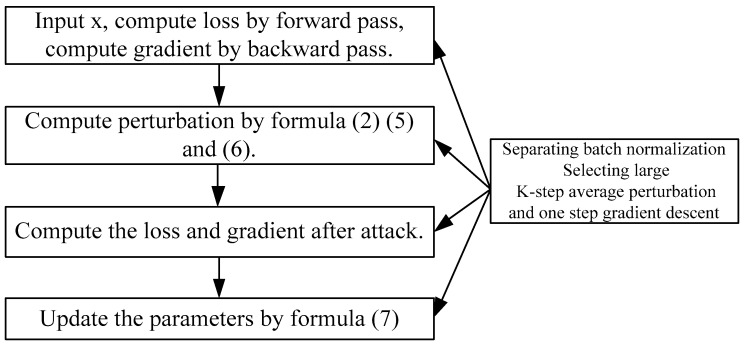
The overall architecture of the proposed method.

**Figure 2 sensors-24-04154-f002:**
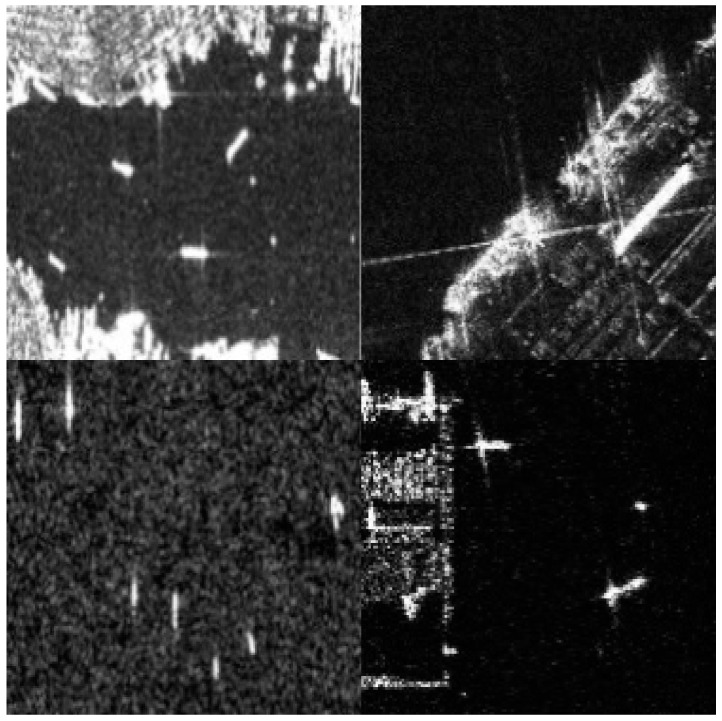
Samples of SSDD.

**Figure 3 sensors-24-04154-f003:**
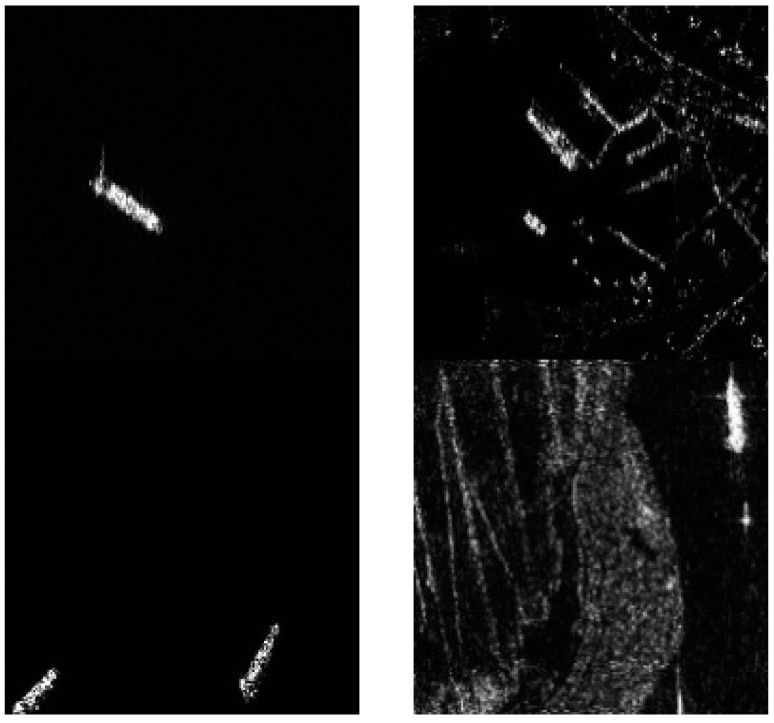
Samples of SAR-Ship-Dataset.

**Figure 4 sensors-24-04154-f004:**
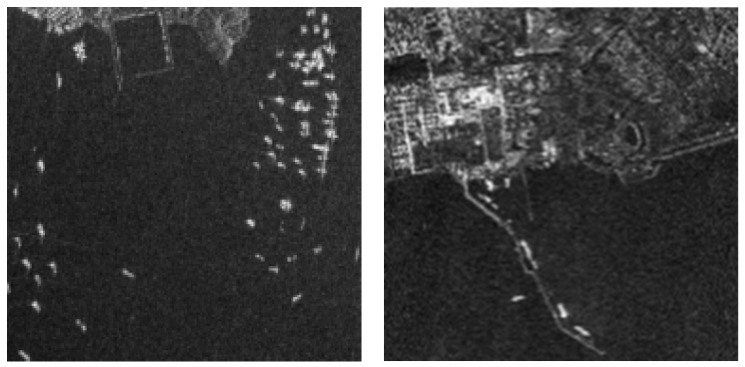
Samples of AIR-SARShip.

**Figure 5 sensors-24-04154-f005:**
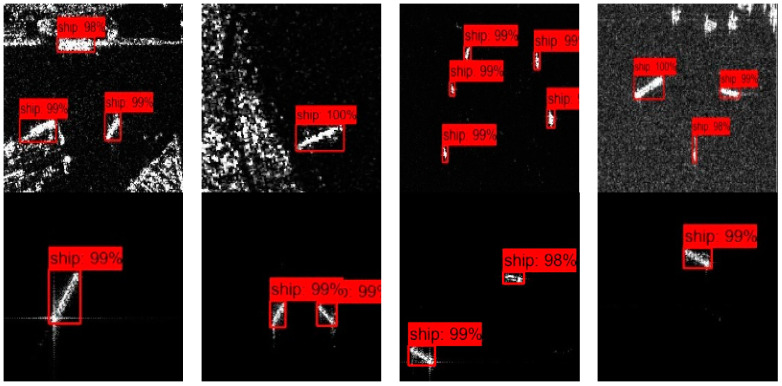
The detection results on the datasets.

**Table 1 sensors-24-04154-t001:** The baselines of the build-in data augmentation methods.

Datasets	FR	RN	Y3	CN	AVE
**SSDD**	0.84	0.90	0.87	0.94	0.89
**SAR-Ship-Dataset**	0.86	0.89	0.79	0.86	0.85
**AIR-SARShip**	0.82	0.86	0.46	0.82	0.74

**Table 2 sensors-24-04154-t002:** The results of different data augmentation methods and detectors on SSDD.

DAM	FR	RN	Y3	CN	AVE1	AVE2
Cutout	0.91	0.93	0.75	0.88	0.87	0.85
Random erasing	0.91	0.91	0.78	0.86	0.87	
Mixup	0.91	0.97	0.82	0.92	0.91	
Cutmix	0.73	0.77	0.71	0.72	0.73	
Augmix	0.92	0.94	0.85	0.91	0.91	
Gridmask	0.90	0.90	0.84	0.85	0.87	
Mosic	0.88	0.89	0.80	0.87	0.86	
Copy/paste	0.82	0.91	0.65	0.88	0.82	
Proposed	0.94	0.95	0.89	0.95	0.93	0.93

**Table 3 sensors-24-04154-t003:** The results of different data augmentation methods and detectors on SAR-Ship-Dataset.

DAM	FR	RN	Y3	CN	AVE1	AVE2
Cutout	0.87	0.91	0.81	0.89	0.87	0.83
Random erasing	0.86	0.88	0.84	0.83	0.85	
Mixup	0.89	0.93	0.80	0.90	0.88	
Cutmix	0.71	0.92	0.52	0.60	0.69	
Augmix	0.90	0.91	0.65	0.88	0.84	
Gridmask	0.85	0.89	0.78	0.84	0.84	
Mosic	0.88	0.86	0.84	0.86	0.86	
Copy/paste	0.86	0.90	0.64	0.89	0.82	
Proposed	0.93	0.94	0.89	0.94	0.93	0.93

**Table 4 sensors-24-04154-t004:** The results of different data augmentation methods and detectors on AIR-SARShip.

DAM	FR	RN	Y3	CN	AVE1	AVE2
Cutout	0.91	0.88	0.40	0.81	0.75	0.72
Random erasing	0.91	0.85	0.42	0.84	0.76	
Mixup	0.90	0.81	0.38	0.86	0.74	
Cutmix	0.62	0.51	0.30	0.83	0.57	
Augmix	0.90	0.86	0.41	0.89	0.77	
Gridmask	0.81	0.76	0.2	0.84	0.65	
Mosic	0.80	0.84	0.65	0.85	0.79	
Copy/paste	0.81	0.78	0.48	0.81	0.72	
Proposed	0.95	0.93	0.73	0.93	0.89	0.89

**Table 5 sensors-24-04154-t005:** The ablation experiment results.

Methods	SSDD	SAR-Ship-Dataset	AIR-SARShip
Traditional methods	0.85	0.83	0.72
Plain adversarial training	0.88	0.87	0.81
+Separate BN	0.89	0.89	0.84
+Maxing	0.91	0.90	0.86
+Optimization	0.93	0.93	0.89

## Data Availability

Dataset available on request from the authors.

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
