# Peer review of "A SAR Ship Detection Method Based on Adversarial Training"

_sensors, 2024, doi:10.3390/s24134154_

Round 1
Reviewer 1 Report
Comments and Suggestions for Authors
1. In line 64, the author proposes in the first contribution that he is the first to apply adversarial learning to SAR ship detection. But in the paper "B. Hou et al., "A Neural Network Based on Consistency Learning and Adversarial Learning for Semisupervised Synthetic Aperture Radar Ship Detection," in IEEE Transactions on Geoscience and Remote Sensing, vol. 60, pp. 1-16, 2022, Art no. 5220816, doi: 10.1109 / TGRS. 2022.3142017 ", against learning have been applied to SAR ship detection. The description here should be changed.
2. In the experimental part, the author only uses AP50 as the evaluation index, and suggests adding evaluation indexes such as: , etc.
3. Figure 5 is too blurry and the confidence value of the detection box cannot be clearly seen.
4. In Table 2, the header should be "The results of different data augmentation methods and detectors on SSDD", instead of "The results of the proposed method on SSDD" because experiments with other data enhancement methods are also performed in the table.
5. In Table 3, for the combination of four detection methods and eight traditional data enhancement methods, the accuracy rate obtained was averaged. Each row of data should be averaged to obtain the average accuracy rate of different methods under each traditional data enhancement method, and then compared with the method proposed in the paper, it is more convincing.
6. Pseudocode is used to illustrate the proposed method.
Comments on the Quality of English Language
Minor editing of English language required
Author Response
Author's Reply to the Review Report (Reviewer 1, second review)
Comments:
Thank you for re-submitting your manuscript. However, while reviewing the revised version, I noticed that many of the suggestions and comments I made during the initial review were not fully addressed. It appears that these revisions were done in a rather cursory manner.
To improve the quality and clarity of the paper, I strongly recommend that you thoroughly consider and incorporate the feedback I provided during the initial review. Addressing these issues will greatly enhance the overall impact and readability of your work.
Please take the necessary time to comprehensively revise your manuscript. I look forward to reviewing a more polished version of your paper.
Response:
The reviewer 1 is likely not satisfied with the answers to the second and sixth questions. As the previous response, we believe that it is not necessary to make further revisions to the paper according to the comments of Reviewer 1. The reasons are as follows.
- For question 2, the AP50 used in the paper can already measure the performance of the detection algorithm. If other indicators are added, not only will it significantly increase the length of the paper, but we will also need to repeat a lot of experiments (108), which is obviously not cost-effective.
- For question 6, although the paper does not have pseudocode, the flowchart in Figure 1 serves the same purpose as pseudocode. So we believe it is not necessary to add pseudocode again.

Reviewer 2 Report
Comments and Suggestions for Authors
The authors proposed a method to detect ship using adversarial training. The language reads smooth. However, the method of the manuscript should be descripted and compared deeply.
Suggestions are as follows,
1. Please provide some quantitative description in the abstract part.
2. Page 2, Line 73, I thinks it should be “…, and use them to update…”.
3. Abbreviations are typically defined the first time the term is used within the abstract and again in the main text and then used exclusively throughout the remainder of the document. Please consider adhering to this convention. The target journal may have a list of abbreviations that are considered common enough that they do not need to be defined.
4. What is the purpose of the authors to use three kinds of data? I guess the authors want to validate the proposed method using the three data sources. However, this is duplicated in the manuscript.
5. What kind of perturbation is used to degrade the images? Please descript the details and give some comparison.
6. The satellite-borne or airborne SAR images should be used to test the method.
Some of the sentences should be modfied. I suggest the authors ask a native speaker for help.
Round 2
Reviewer 1 Report
Comments and Suggestions for Authors
Thank you for re-submitting your manuscript. However, while reviewing the revised version, I noticed that many of the suggestions and comments I made during the initial review were not fully addressed. It appears that these revisions were done in a rather cursory manner.
To improve the quality and clarity of the paper, I strongly recommend that you thoroughly consider and incorporate the feedback I provided during the initial review. Addressing these issues will greatly enhance the overall impact and readability of your work.
Please take the necessary time to comprehensively revise your manuscript. I look forward to reviewing a more polished version of your paper.
Comments on the Quality of English LanguageThe quality of English language in your manuscript needs improvement to meet the journal's standards. I suggest seeking help from a native English speaker or a professional language editing service to enhance the clarity and readability of the paper. This will ensure that your research is communicated more effectively to readers.
Reviewer 2 Report
Comments and Suggestions for Authors
My questions have been answered, I have no other questions.